# Ultrasound Versus Computed Tomography for Diaphragmatic Thickness and Skeletal Muscle Index during Mechanical Ventilation

**DOI:** 10.3390/diagnostics12112890

**Published:** 2022-11-21

**Authors:** Stefano Gatti, Chiara Abbruzzese, Davide Ippolito, Sophie Lombardi, Andrea De Vito, Davide Gandola, Veronica Meroni, Vittoria Ludovica Sala, Sandro Sironi, Antonio Pesenti, Giuseppe Foti, Emanuele Rezoagli, Giacomo Bellani

**Affiliations:** 1Department of Emergency Medicine, ASST Monza, San Gerardo Hospital, Via Pergolesi 33, 20900 Monza, Italy; 2School of Medicine and Surgery, University of Milano-Bicocca, Via Cadore 48, 20900 Monza, Italy; 3Department of Anesthesia, Critical Care and Emergency, Fondazione IRCCS Ca’ Granda Ospedale Maggiore Policlinico, Via Francesco Sforza 28, 20122 Milan, Italy; 4Department of Diagnostic Radiology, H. San Gerardo, Via Pergolesi 33, 20900 Monza, Italy; 5Department of Diagnostic Radiology, ASST Papa Giovanni XXIII, 24127 Bergamo, Italy; 6Department of Pathophysiology and Transplantation, University of Milan, Via Festa del Perdono 7, 20122 Milan, Italy

**Keywords:** ultrasound, mechanical ventilation, computed tomography, diaphragm, skeletal muscle index, critical care

## Abstract

***Background:*** Diaphragmatic alterations occurring during mechanical ventilation (MV) can be monitored using ultrasound (US). The performance of computed tomography (CT) to evaluate diaphragmatic thickness is limited. Further, the association between muscle mass and outcome is increasingly recognized. However, no data are available on its correlation with diaphragmatic thickness. We aimed to determine correlation and agreement of diaphragmatic thickness between CT and US; and its association with muscle mass and MV parameters. ***Methods:*** Prospective observational study. US measurements of the diaphragmatic thickness were collected in patients undergoing MV within 12 h before or after performing a CT scan of the thorax and/or upper abdomen. Data on skeletal muscle index (SMI), baseline, and ventilatory data were recorded and correlated with US and CT measures of diaphragmatic thickness. Agreement was explored between US and CT data. ***Results***: Twenty-nine patients were enrolled and the diaphragm measured by CT resulted overall thicker than US-based measurement of the right hemidiaphragm. The US thickness showed the strongest correlation with the left posterior pillar at CT (r = 0.49, *p* = 0.008). The duration of the controlled MV was negatively correlated with US thickness (r = −0.45, *p* = 0.017), the thickness of the right anterior pillar (r = −0.41, *p* = 0.029), and splenic dome by CT (r = −0.43, *p* = 0.023). SMI was positively correlated with US diaphragmatic thickness (r = 0.50, *p* = 0.007) and inversely correlated with the duration of MV before enrollment (r = −0.426, *p* = 0.027). ***Conclusions:*** CT scan of the left posterior pillar can estimate diaphragmatic thickness and is moderately correlated with US measurements. Both techniques show that diaphragm thickness decreases with MV duration. The diaphragmatic thickness by US showed a good correlation with SMI.

## 1. Introduction

Muscle wasting and dysfunction is a frequent problem in the intensive care unit (ICU) [1]. There is an association between diaphragmatic inactivity and development of atrophy and dysfunction [2,3]. Impaired function of respiratory muscles, particularly of the diaphragm, during mechanical ventilation (MV) represents a relevant clinical issue because it may lead to prolonged ventilation and difficult weaning [4,5]. The concept of ventilator-induced diaphragmatic dysfunction has been described; based on some data, it could be aggravated by administration of neuromuscular blockers [6] and high doses of corticosteroids [7], even if, in this respect, controversial data exist [8,9].

In clinical practice, reliable tools to monitor diaphragmatic structure and function are needed. These tools might become useful as intermediate end-points in future clinical trials, aimed at developing disease-modifying interventions.

Therefore, several tests have been proposed, some techniques rely on global estimation of respiratory muscle strength [10], trans-diaphragmatic pressure generation, following transdermal phrenic nerve magnetic stimulation [11,12], or assessment of the electrical activity amplitude of the diaphragm (EAdi) [13]. Recent findings showed that ultrasound (US) examination of the diaphragm is useful for monitoring its structure and function during MV through the evaluation of diaphragmatic thickness and thickening [14,15,16].

During US imaging of the transthoracic diaphragm, the diaphragmatic thickness is measured at the ninth or tenth intercostal space, near the midaxillary line across the zone of apposition [14,17]. However, whether these measurements represent the whole diaphragm is still unclear; it is an interesting issue because the diaphragm thickness has been described to be heterogeneous across its surface [18].

Although the computed tomography (CT) scan has been extensively applied to lung field evaluations [19], few studies focused on its potential to evaluate diaphragmatic thickness. Compared with US, CT scanning has several disadvantages, such as the need to transfer the patient, X-ray exposure, lower spatial resolution, and contrast with surrounding structures. At the same time, CT scanning offers the advantage of imaging several portions of the diaphragm. Moreover, the diaphragm is normally visualized by CT scans of the thorax and upper abdomen performed for clinical purposes, so additional X-rays are not required.

There is increasing interest in assessing sarcopenia in critical settings since patients in the ICU are at increased risk of muscle weakness associated with declining muscle mass and function [20,21]. Patients affected by critical illness-associated sarcopenia are at increased risk of mortality, longer hospital admission, and higher readmission rate. Presence of sarcopenia during hospital stay may lead to loss of functional independence after discharge [22]. Moreover, a correlation between a single CT scan derived index (skeletal muscle index—SMI) and a worse outcome in terms of survival among surgical patients admitted to ICU was demonstrated in recent studies [23,24,25]. The CT scan is a validated technique in assessing lean body mass at the level of lumbar vertebrae [26,27], however, recent findings suggest the relevance of the thoracic CT scan in assessing sarcopenia [28].

In this study, we aimed to determine the agreement and correlation between CT scan measurements in different portions of the diaphragm and US measurements of diaphragmatic thickness, considering the CT scan as an objective and non-operator dependent technique that can describe the entire diaphragm. We evaluated the relationship between diaphragmatic thickness and clinical characteristics and outcomes of ICU patients undergoing MV. Furthermore, we aimed to stratify our population according to different levels of lean body mass calculated by the thorax CT scan and to explore if there was association with diaphragmatic thickness.

## 2. Materials and Methods

This prospective observational study was performed in the general ICUs of San Gerardo Hospital in Monza and of Fondazione IRCCS Ca’ Granda Ospedale Maggiore Policlinico in Milan, Italy, between February and December 2016. The study was approved by the Ethics Committee of the hospitals, and all patients or their substitute decision makers provided informed consent. Intubated or tracheotomized patients undergoing MV for at least 48 h who also underwent a CT scan of the thorax and/or upper abdomen for clinical reasons were considered eligible. Patients younger than 18 years and those with pre-existing neuromuscular diseases, phrenic nerve lesions, or air leakage were excluded from the study.

Demographic information and clinical data (comorbidity, admission diagnosis, blood tests, drug therapy, arterial blood gas analysis, and ventilatory data) were collected during enrollment. As a “control” group, we used a sample of 16 subjects on spontaneous breathing, without any form of ventilatory assistance, who underwent a CT scan of the thorax and/or upper abdomen at San Gerardo Hospital in Monza and whose data were stored in our radiological database. This study was conducted according to the guidelines STrengthening the Reporting of OBservational studies in Epidemiology (STROBE) [29].

### 2.1. CT Protocol and Image Analysis

CT scans were performed based on standard hospital protocol and executed with two different multidetector CT scanners (Brilliance CT 16-slice scanner and Brilliance iCT 256-slice scanner; Philips Healthcare, Best, The Netherlands). The acquired volumes were included between the apices and bases of the lung with the following parameters: pitch 0.9 to 1.1; slice thickness of 2 mm; 120 kV with automated tube current modulation (range, 120–350 mA). CT scans were performed with or without contrast administration based on clinical needs. Based on local protocols, all patients were under controlled MV during image acquisition; neuromuscular blockers were administered to facilitate the patients’ transfer. The mean acquisition scan time was 2.3 ± 0.5 s and 2.8 ± 0.4 s for the study and control groups, respectively, with no statistical difference between the two groups. The diaphragmatic thickness was measured by a radiologist (D.I) with ten years of experience in advanced radiological imaging, who was blinded to clinical and US findings, with the support of a workstation (AGFA Diagnostic Software, Impax, version 6.4.0.3125; Agfa, Mortsel; Belgium). The diaphragm’s maximum thickness (expressed in mm) and density (expressed in Hounsfield unit) were evaluated in six different areas: (a) right anterior pillar, (b) left anterior pillar, (c) right posterior pillar, (d) left posterior pillar, (e) hepatic, and (f) splenic domes. The pillars represent the different portions of the diaphragm (the lumbar diaphragm or crura and costal diaphragm), and they are recognizable as bundles of muscular fibers: two arising from the anterolateral surface of the first three right lumbar vertebrae, called “left and right crus of the diaphragm” (where median and lateral arcuate ligaments arise), two anterior parts (between the xiphoid and the middle leaflets of the central diaphragmatic tendon) covering the dome of the liver and spleen, and two costal parts, covering the lateral part of the liver and spleen. CT provides an overall precise image of the different pillar components. For each different area, the radiologist performed three different measurements of the diaphragm thickness in three different sections, avoiding distortion or artifact regions. The mean of the three measurements was calculated (Figure 1). A free-hand region of interest (ROI) was drawn at the same level of thickness measurements to calculate tissue density in terms of the absolute Hounsfield unit.

Moreover, an axial single-slice cross-sectional image at the 12th dorsal vertebra level was analyzed by ImageJ (developed by the National Institutes of Health; available from https://imagej.nih.gov/ij/download.html, accessed on 1 December 2016), as follow. Total muscle area (TMA), which included erector spinae, latissimus dorsi, external and internal oblique, rectus abdominis, and external and internal intercostal muscles, was calculated automatically by using muscle tissue pixel density thresholds (between −29 and +150 Hounsfield-Unit) after radiology had drawn a ROI inside and outside muscular profiles. TMA values were normalized by height calculating the skeletal muscle index (SMI) according to the formula SMI = TMA/m^2^ [28].

### 2.2. US Protocol and Image Analysis

Ultrasonographic measurements of diaphragmatic thickness were collected within the 12-h period before or after the CT scan was performed. All US scans were performed using the same US system at each institution (MyLabTM25Gold, 12 MHz linear probe LA523, Esaote, Genova, Italy; iU22 xMatrix, 2–17 mHz linear array, Philips, Amsterdam, Netherlands). The linear transducer was placed in the ninth or tenth intercostal space near the midaxillary line and angled perpendicular to the chest wall as described by Goligher et al. [14,17]. At this level, the diaphragm is identified as a three-layered structure just superficial to the liver, consisting of a relatively non-echogenic muscular layer bound by the echogenic membranes of the diaphragmatic pleura and peritoneum [14] (Appendix A). Diaphragmatic thickness was measured at the end expiration, and two different measurements were performed for each hemidiaphragm. The mean of the two measurements was used for subsequent statistical analysis. Diaphragmatic US was performed by experienced ICU physicians after a training period, during which good between-investigator reproducibility was established [14].

## 3. Statistical Analysis

Descriptive statistics included mean (standard deviation) or median (range interquartile) for continuous variables and proportion for categorical variables. The Shapiro–Wilk test was used to evaluate normality of data distribution. In each subject, the variation in measures obtained by CT scan was quantified using the coefficient of variation (CoV). CoV was calculated as the SD of the average CT diaphragmatic thicknesses evaluated over the six diaphragm areas and then normalized by the mean. Differences between continuous variables were tested by unpaired Student’s *t*-test or Mann–Whitney U test as appropriate. Correlations between variables were evaluated using the Pearson correlation coefficient. Analyses of agreement between CT scan and US was performed by using the Bland–Altman analysis. Bias with 95% confidence interval (CI) was reported.

All statistical analyses were two-sided, and *p* < 0.05 was required for statistical significance. Analyses were conducted using SPSS Statistics^®^ version 27.0 (IBM Corporation, Armonk, NY, USA), STATA software version 16/MP (StataCorp LP, College Station, TX, USA), and GraphPad Prism 7.a (GraphPad Software, San Diego, CA, USA).

## 4. Sample Size

Based on previous studies, we would like to detect a correlation slope of at least 0.6 between diaphragmatic thickness measured with CT scan and US, statistical difference from 0 with a type 1 error of 0.05 and a power of 0.80. Based on this assumption, we estimated a sample size of 24 patients. Moreover, we planned to enroll at least 27 patients if diaphragmatic thickness was not evaluable for 10% of patients.

## 5. Results

### 5.1. Study Population

We enrolled 29 consecutive patients who underwent MV, including 22 men and 7 women. No patients were excluded, and measurement of diaphragmatic thickness through CT scan was feasible in all patients. US evaluation of left hemidiaphragm could not be obtained consistently, therefore we considered only right US measurements. Demographic and clinical characteristics of enrolled patients are reported in Appendix A.

### 5.2. Coherence between Imaging Techniques

Diaphragm thickness was heterogeneous among different regions analyzed by CT scan, such that left anterior and posterior pillar, right posterior pillar and splenic dome, as well as mean diaphragmatic thickness were significantly thicker than the US-based measurement of the right hemidiaphragm (Figure 2).

The thickness measurements, which were obtained using a CT scan in different regions of the diaphragm, were significantly correlated with each other (Pearson correlation coefficient (r) varying from 0.376 to 0.779), and single Pearson correlation coefficient was reported in the Appendix A. The highest correlation was found between the right and left posterior pillar measurements (r = 0.779; *p* < 0.001).

Correlation coefficients for diaphragmatic thickness obtained using US and CT scan are shown in Table 1. We found no significant correlation between the mean diaphragmatic thickness measured using CT scan and US (r = 0.343, *p* = 0.074), in contrast, we did find a significant correlation at a regional level between US and CT scan measurements at the left posterior pillar (r = 0.488, *p* = 0.008). Bland–Altman plots between US and CT scan are reported in Figure 3 and Appendix A, highlighting limited agreement between the techniques as observed by the wide limits of agreement despite a low bias.

### 5.3. Correlation between Diaphragmatic Thickness and Duration of Mechanical Ventilation

The Pearson correlation coefficients between the controlled, assisted, and total duration of MV and thickness by CT and US are reported in Table 2. The duration of controlled ventilation before enrollment was negatively correlated with US thickness (r = −0.449, *p* = 0.017) (Figure 4C). Considering the CT scan, the relationship between thickness by CT and controlled ventilation before enrollment was statistically significant at the level of the right anterior and splenic dome, as reported in Table 2.

### 5.4. Sarcopenia, Diaphragmatic Thickness, and Mechanical Ventilation

US diaphragmatic thickness was positively correlated with the SMI value (r = 0.496—*p* = 0.007, Figure 4B) and SMI value was inversely correlated with the duration of mechanical ventilation before enrollment (r = -0.426—*p* = 0.027, Figure 4A). On the contrary, there is no correlation between SMI and thickness measured considering the CT scan in the different areas of the diaphragm (Appendix A).

Subsequently, we evaluated the study population by stratifying SMI into two groups using the median value: high SMI (>37 cm^2^/m^2^) and low SMI (≤37 cm^2^/m^2^). Patients with low SMI were older than patients with high SMI (61 ± 13 years vs. 50± 13 years, *p* = 0.028) (Appendix A). Evaluating measurements performed with US, diaphragmatic thickness was smaller in the low SMI group (2.1 ± 0.4 mm vs. 2.7 ± 0.5 mm, *p* < 0.001) (Appendix A). However, we observed no difference in diaphragmatic thickness measured with CT scan between the two groups.

### 5.5. Comparison between ICU Patients and Healty Patients

In the comparison group, we used a population of non-ventilated patients who underwent CT scans of the thorax or of the upper abdomen for clinical reasons (16 patients, including eight men and eight women, with a mean age of 52 years). Diaphragmatic thickness did not differ across every area of the diaphragm. Moreover, we calculated the CoV between all diaphragmatic areas in each patient. We found that CoV in ventilated patients was not different from CoV in healthy individuals (CoV healthy patients = 0.126; CoV ICU patients = 0.167; *p* = 0.102).

## 6. Discussion

The main findings of this study can be summarized as follows:In a mixed population of patients undergoing MV, the diaphragmatic thickness measured by CT scan in the left posterior pillars showed a moderate correlation with the measurements obtained using US;The diaphragmatic thickness evaluated by US was negatively correlated with the duration of controlled MV before enrollment while the mean diaphragmatic thickness showed a trend towards significance;The diaphragmatic thickness measured by US was moderately correlated with sarcopenia measured by CT scan and the smaller the thickness of the diaphragm was, the lower SMI was, a parameter that may suggest the presence of sarcopenia.

US is becoming a standard technique to measure diaphragm thickness and contraction during MV. The technique, however, can image only one part of the diaphragm, namely the juxtaposition zone of the right costal hemidiaphragm. In this respect, whether this portion is representative of the entire muscle is unknown, particularly considering that the diaphragm is composed of two separate parts (costal and crural). In contrast, the CT scan offers the clear advantage of imaging the entire diaphragm. However, it has different and relevant limitations, such as a lower spatial resolution and a more difficult visualization of the diaphragm versus the surrounding structures. Indeed, reports on CT-based measurements of the diaphragm are quite scarce, mainly focused on evaluating diaphragmatic injury [30,31], and usually restricted to the area of the diaphragmatic crura. Jung et al. measured the diaphragm volume in a retrospective series of CT scans from septic and non-septic patients. Compared with other muscles, they showed that the diaphragm is more susceptible to atrophy. Moreover, they showed that the diaphragm volume is correlated with tracheal pressure under magnetic phrenic stimulation [32]. This study suggested, for the first time, the feasibility of quantitating diaphragm mass by CT; however, the measurements were obtained by a specific three-dimensional analysis software, and external validation was not performed.

To the best of our knowledge, the present study is the first attempt to evaluate the feasibility of measuring diaphragmatic thickness in patients on MV who underwent a CT scan. We found that the diaphragm thickness is heterogeneous along the surface [18], and the posterior pillars, belonging to the crural part, are thicker. However, the coefficient of variance of diaphragm thickness in our patients was not different compared to non-ventilated patients. This result warrants further scrutiny, and this study was not powered to establish whether thickness heterogeneity is a pathological marker of worse dysfunction. Although we do not have a “reference” technique to assess CT-derived measurements’ reliability, we believe that our results support the evidence that diaphragmatic thickness can be adequately measured by CT scan with more reliable results in the posterior pillars’ region. Perhaps this region of the diaphragm can be better assessed and measured with CT in relation to its clear delineation from surrounding structures (i.e., paravertebral muscles, fat tissue) (Figure 1).

Our findings confirm the results of previous studies [30] showing that the anterior or costal part of the diaphragm is usually more difficult to identify than the lumbar part on transverse images. Indeed, the different anterior appearances depend on the cephalocaudal relationship between the xiphoid and the middle leaflets of the central diaphragmatic tendon, making a correct evaluation challenging in most cases.

The reliability of CT-based measurements of the left posterior pillar regions is supported by the consistency measurement with the contralateral pillar and the tight correlation with the US-derived measurement. The fact that the correlation slope is higher than one probably reflects the two structures’ different anatomies and implies that the CT-derived measurement of the left posterior pillar cannot be compared directly to US-derived measurements. Moreover, we also found that the left posterior pillar has a similar relationship to US regarding the alteration of diaphragm thickness associated with MV. The diaphragm has been known to become thinner in many cases during MV15. We replicated this finding using US, but the measurements obtained using CT showed a very similar negative trend of correlation with the duration of ventilation before imaging.

In our opinion, these results should be considered relevant both for clinical and research fields because they are consistent with previous literature findings on US. We proposed and validated a comparable method to quantify diaphragmatic thickness by CT which can be used in the future. US is simpler than CT scan and the last methods could be preferred in some situations, such as further expanding the consequences of diaphragmatic heterogeneity on patient outcomes, or to assess the effect of therapeutic strategies on different diaphragmatic regions. Moreover, large CT scan datasets could be retrospectively analyzed using this method. Likewise, we reinforce the validity of US measurements, which, despite being obtained in a single location, are useful markers of the entire diaphragmatic mass in the critically ill.

Finally, we observed that diaphragm thickness by US was different between patients that were stratified by higher or lower levels of SMI, while no difference was reported by measuring the diaphragm by CT scan. Furthermore, we described a moderate association between the levels of SMI and the diaphragm thickness measured by US. Both variables showed a moderate correlation with the days spent on mechanical ventilation, suggesting that the muscle mass might decrease over prolonged days of mechanical ventilation. The measurement of SMI in critically ill patients was recently suggested to be clinically relevant for its association with outcome [33]. The measurement of sarcopenia by diaphragm US was reported to be feasible and helpful to diagnose sarcopenia in patients with a chronic disease such as sarcoidosis [34]. Furthermore, the US evaluation of the diaphragm thickness was found to independently correlate to the sarcopenic status in patients over 65 y/o regardless the spontaneous breathing modality [35]. Our findings suggest that the phenomenon of muscle wasting may be assessed by evaluating the diaphragm thickness non-invasively by using US. Its validation would be important to assess the presence of sarcopenia in critically ill patients at bedside.

This study has some limitations. We studied a heterogeneous population of patients undergoing MV at different times during their ICU stay. The indications for CT scan are very heterogeneous, but certainly do not include evaluating diaphragmatic mass or thickness. For this reason, we do not believe this represents a selection bias. The acquisition timing could not be standardized because we took advantage of the CT scan performed for clinical indications. US evaluation was performed 12 h previous or after CT scan. During this period, diaphragmatic thickness may modify, even if, according to previous studies [15], this change usually happen over days. Some discrepancies in terms of diaphragm thickness between CT and US findings may be explained by muscle contraction variation that occurs during ventilation, whereas we standardized the measurement during expiration with US, and CT scan was continuously acquired during the respiratory cycle. However, in our clinical practice, patients undergo controlled ventilation with administration of neuromuscular blockers during radiological imaging: the diaphragmatic thickness is rather constant throughout the respiratory cycle when there is no active effort [14].

In conclusion, this study shows that a CT scan estimate of the thickness of the left posterior pillar of the crural portion of the diaphragm moderately correlates with the thickness of the juxtaposition zone of the costal part measured using US; the measurements obtained by the two methods showed a similar behavior in relation to the MV duration. Both these techniques show that diaphragm thickness decreases with ventilation duration.

## Figures and Tables

**Figure 1 diagnostics-12-02890-f001:**
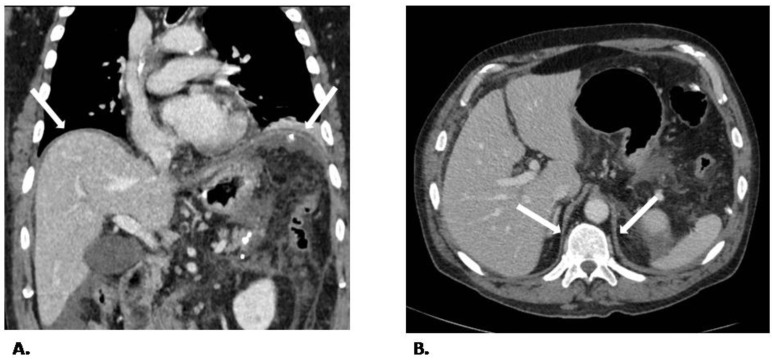
Normal appearance of the diaphragm. The computed tomography scan clearly shows the crura in the direct coronal (**A**) and axial planes (**B**). The normal appearance of the diaphragm, both in the posterior region (crura) at the liver dome level and the left anterior diaphragm (arrows).

**Figure 2 diagnostics-12-02890-f002:**
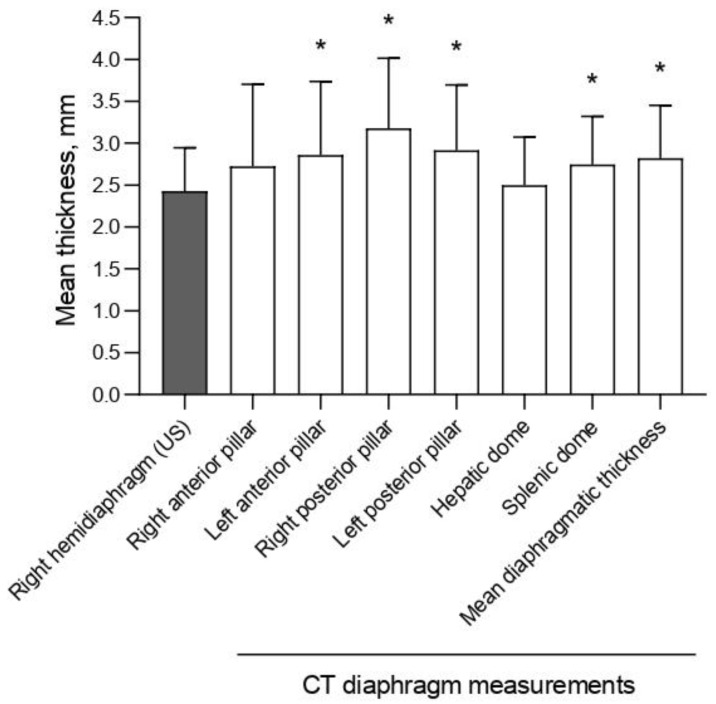
The mean diaphragmatic thickness measured using ultrasound (US right hemidiaphragm) and computed tomography (CT, at the level of different areas of the diaphragm). CT scan diaphragmatic thickness at the level of left anterior, right posterior, left posterior pillars, and splenic dome is significantly thicker than US-based thickness. * *p* < 0.05 (two-tailed) versus US-based thickness.

**Figure 3 diagnostics-12-02890-f003:**
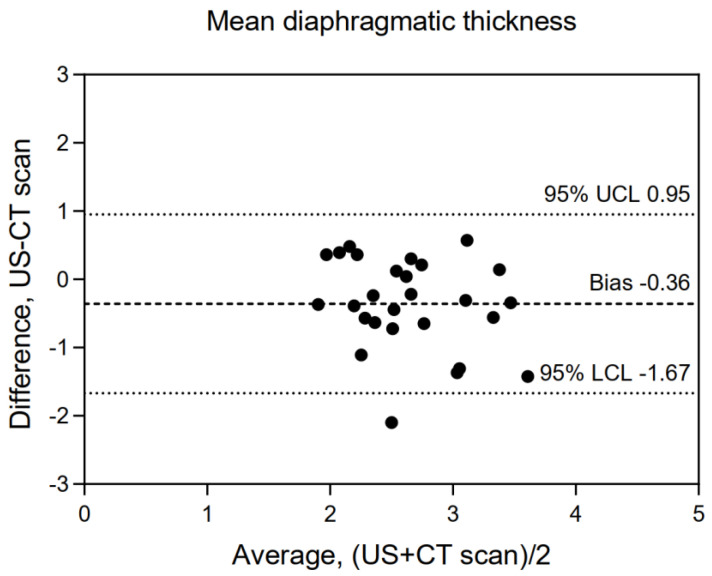
Bland–Altman plot exploring the agreement between the right hemidiaphragm thickness evaluated by US and the mean diaphragmatic thickness assessed by CT scan. Data are expressed in mm. UCL = upper confidence limit; LCL = lower confidence limit. Definition of abbreviation: US = ultrasound; CT scan = computerized tomographic scan.

**Figure 4 diagnostics-12-02890-f004:**
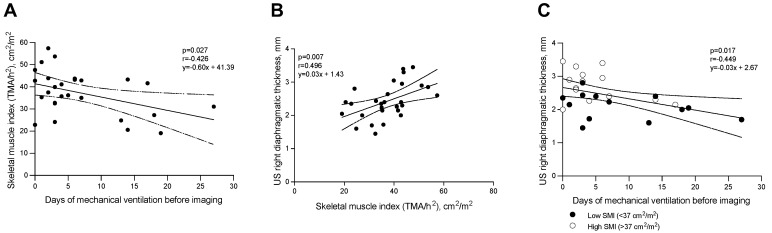
Correlation between days of mechanical ventilation (MV) before imaging and Skeletal Muscle Index (SMI) (**A**); between SMI and US right diaphragmatic thickness (**B**); and days of MV before imaging and US right diaphragmatic thickness (**C**). Definition of abbreviation: SMI = skeletal muscle index; TMA = total muscle area; US = ultrasound.

**Table 1 diagnostics-12-02890-t001:** Correlation between computerized tomography and ultrasound measurements of diaphragmatic thickness.

Diaphragmatic Areas Sampled by CT Scan	Mean Thickness (mm)	Correlation with US Thickness (r)	
		r	*p*-Value
Right anterior pillar	2.73 ± 0.98	0.346	0.072
Right posterior pillar	3.18 ± 0.84	0.310	0.108
Hepatic dome	2.50 ± 0.57	0.163	0.408
Left anterior pillar	2.86 ± 0.88	0.081	0.682
Left posterior pillar	2.91 ± 0.78	0.488	0.008
Splenic dome	2.75 ± 0.57	0.287	0.139
Mean diaphragmatic thickness	2.82 ± 0.63	0.343	0.074
US right hemidiaphragm	2.44 ± 0.52	/	/

In Table 1 the mean thickness measured at the level of different areas of the diaphragm by CT scan, Pearson correlation coefficients (r) and *p*-values between diaphragmatic thickness measured with US, and thickness of single diaphragmatic areas measured with CT scan are reported. Definition of abbreviation: US = ultrasound; CT scan = computerized tomographic scan.

**Table 2 diagnostics-12-02890-t002:** Correlation between diaphragmatic thickness and duration of mechanical ventilation until enrollment.

	Duration of MV					
	Total		Controlled		Assisted	
	**r**	** *p* ** **-Value**	**r**	** *p* ** **-Value**	**r**	** *p* ** **-Value**
CT scan right anterior pillar	−0.332	0.084	−0.412	0.029	−0.033	0.868
CT scan right posterior pillar	0	0.984	−0.111	0.573	0.140	0.477
CT scan hepatic dome	0	0.972	−0.184	0.349	0.231	0.238
CT scan heft anterior pillar	−0.118	0.550	−0.246	0.208	0.120	0.544
CT scan left posterior pillar	−0.184	0.348	−0.358	0.061	0.153	0.437
CT scan splenic dome	−0.349	0.069	−0.428	0.023	−0.040	0.838
CT scan mean diaphragmatic thickness	−0.206	0.294	−0.353	0.065	0.109	0.580
US thickness	−0.371	0.047	−0.449	0.017	−0.063	0.745

In Table 2 the Pearson correlation coefficient (r), 95% confidence interval and p-value between thickness of single diaphragmatic areas measured with CT scan and ventilation features are reported. Definition of abbreviation: CT scan = computerized tomographic scan, US = ultrasound, MV = mechanical ventilation.

## Data Availability

Data are available upon reasonable request to the corresponding author.

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
