# Peer review of "Ultrasound Versus Computed Tomography for Diaphragmatic Thickness and Skeletal Muscle Index during Mechanical Ventilation"

_diagnostics, 2022, doi:10.3390/diagnostics12112890_

Round 1

Reviewer 1 Report

General comments

The paper is well-written and I enjoyed reading it.  The topic is of uncertain importance, but with growing use of ultrasound scanning by intensive care doctors, and the ongoing scientific interest in mechanical ventilation, the paper is likely to attract some interest from readers, especially in the future.  The purpose of the study is a little obscured by its complexity and the multiplicity of analyses.  The final result and conclusion could also be expressed more clearly for the casual reader.  What are we conclude (generally) from the fact that the CT scan measurements are ‘moderately’ correlated with ultrasound measurements?  Does this point to clinical utility of this technique, in the opinion of the authors?

It is not quite clear to me whether the authors consider CT-scanning of the diaphragm a ‘gold-standard’ against which US is then being compared.  I think the basic premise of the study could be explained more clearly.

I think the authors might consider placing some of the Bland-Altman plots in the supplemental material.

Could the timing of image acquisition (end expiration for US but random for CT) explain the fact that the CT measurements were (on average) slightly higher than for US?  What is known about the effect of respiration phase and diaphragmatic thickness?  There is a comment in Discussion (lines

369-372) that seems  unsupported by published evidence.  I think this line of defence is insufficient without evidence, and greater acceptance of this (perhaps) important limitation is warranted.

 A (perhaps inexpert) reading of the graphic results (Figure 2) would seem to suggest that Right hemidiaphragm US measurement most closely approximates the right anterior pillar CT measurement, or perhaps the hepatic dome.  Yet it is the left posterior pillar CT measurement that seems to be the winner.  Perhaps I am misunderstanding something here, but I think this could be better explained.

Broadly, I think this nice piece of research deserves a place in the published literature.  Removal of some of the analysis to a Supplemental Materials section might make the paper more approachable to the general reader.

Specific comments

Abstract

Number of subjects not stated.

Line 34 ‘recognized’.

Introduction

For a (in my reading) proof-of-concept methodological study, the Introduction is could be shortened a little. 

Materials and Methods

Line 152 ‘…as follow: ….’  (note colon)

Line 156 ‘ROI’ not defined

 Line 155  ‘after radiology had drawn…’

 The design of Table 1 I find slightly confusing.  The last column is labelled as the r values’, but it seems also to contain P-values.  Perhaps this table could be better designed.  The labelling of columns in Table 2 seems different.

Page 8 line 239 ‘With the CT scan (rather than ‘considering the CT scan’)

Page 9 Paragraph concerning stratifying by SMI.  Was this useful?  Perhaps it could be omitted or (simply the analysis) placed in Supplemental materials, for the more enthusiastic reader.

Page 11 Line 346 ‘Finally,…’ (rather than ‘At last…’)

Author Response

Reviewer #1

C1. General comments

The paper is well-written and I enjoyed reading it.  The topic is of uncertain importance, but with growing use of ultrasound scanning by intensive care doctors, and the ongoing scientific interest in mechanical ventilation, the paper is likely to attract some interest from readers, especially in the future.  The purpose of the study is a little obscured by its complexity and the multiplicity of analyses.  The final result and conclusion could also be expressed more clearly for the casual reader.  What are we conclude (generally) from the fact that the CT scan measurements are ‘moderately’ correlated with ultrasound measurements?  Does this point to clinical utility of this technique, in the opinion of the authors?

R1. Many thanks for the positive feedback and constructive observations. The primary aim of the study is to explore the agreement - evaluated through bias and 95% CI by Bland Altman plots - between two different techniques. Secondary objectives aim to investigate: 1. how two different techniques such as CT scan – that is not-operator dependent - and ultrasound (UC) – that, in contrast, is operator dependent - are associated; and 2. how an US-evaluation of a single area of the diaphragm may be representative of the whole structure of the diaphragm. Moreover, we believe that it may be of relevant clinical interest that the diaphragm thickness measured by US is moderately associated with the skeletal muscle index (SMI) and that diaphragmatic thickness is different between patients that were stratified by higher or lower levels of SMI. This may be clinically relevant insight considering that SMI is associated with outcome in literature (doi: 10.1016/j.nut.2022.111687).

In conclusion we think that the strength of the diaphragmatic evaluation by US is the good agreement with the CT scan measurements (which are objective and non-operator dependent) and its association with SMI.

C2. It is not quite clear to me whether the authors consider CT-scanning of the diaphragm a ‘gold-standard’ against which US is then being compared.  I think the basic premise of the study could be explained more clearly.

R2. We considered CT scan as an objective and non-operator dependent technique that can measure diaphragm across its whole surface (REF. 29 of the manuscript: Jung B, Nougaret S, Conseil M, Coisel Y, Futier E, Chanques G, Molinari N, Lacampagne A, Matecki S, Jaber S: Sepsis is associated with a preferential diaphragmatic atrophy: a critically ill patient study using tridimensional computed tomography. Anesthesiology 2014; 120:1182–1191). Our aim was to estimate the reliability of US as an operator dependent technique that explores a limited area of the diaphragm. We now better detailed this information at the end of introduction as follows: “In this study, we aimed to determine the agreement and correlation between CT scan measurements in different portions of the diaphragm and US measurements of diaphragmatic thickness, considering CT scan as an objective and non-operator dependent technique that can describe the entire diaphragm. We evaluated the relationship between diaphragmatic thickness and clinical characteristics and outcomes of ICU patients undergoing MV. Furthermore, we aimed to stratify our population according to different levels of lean body mass calculated by thorax CT scan and to explore if there was association with diaphragmatic thickness.”.

C3. I think the authors might consider placing some of the Bland-Altman plots in the supplemental material.

R3. We now added Bland-Altman plots for single CT areas of the diaphragm in the supplemental materials section.

C4. Could the timing of image acquisition (end expiration for US but random for CT) explain the fact that the CT measurements were (on average) slightly higher than for US?  What is known about the effect of respiration phase and diaphragmatic thickness?  There is a comment in Discussion (lines 369-372) that seems  unsupported by published evidence.  I think this line of defence is insufficient without evidence, and greater acceptance of this (perhaps) important limitation is warranted.

R4. The effect of respiratory phase during mechanical ventilation in patients receiving neuromuscular blockers (as in the case of our patients) is considered negligible. This was demonstrated by Goligher et al. in an paper that we now referenced (Ref 14: Goligher EC, Laghi F, Detsky ME, Farias P, Murray A, Brace D, Brochard LJ, Bolz SS, Rubenfeld GD, Kavanagh BP, Ferguson ND: Measuring diaphragm thickness with ultrasound in mechanically ventilated patients: feasibility, reproducibility and validity. Intensive Care Med 2015; 41:642–649). We apologize for the oversight.

C5. A (perhaps inexpert) reading of the graphic results (Figure 2) would seem to suggest that Right hemidiaphragm US measurement most closely approximates the right anterior pillar CT measurement, or perhaps the hepatic dome.  Yet it is the left posterior pillar CT measurement that seems to be the winner.  Perhaps I am misunderstanding something here, but I think this could be better explained.

R5. We thank the Reviewer for highlighting this point. We apologize if we did not clarify the findings enough. We confirm that the primary aim of our study was the agreement between US and CT measurements that we assessed by the Bland-Altman analysis. The right hemidiaphragm US measurements had an excellent agreement – as reported by the Bland-Altman bias – with the hepatic dome and the right anterior pillar assessed by CT.

About the analysis of association between US and CT measurements, which was among our secondary aims, the highest correlation between right hemidiaphragm US measurements and the CT measurements was with the left posterior pillar and the right anterior pillar (Table 2).

C6. Broadly, I think this nice piece of research deserves a place in the published literature.  Removal of some of the analysis to a Supplemental Materials section might make the paper more approachable to the general reader.

R6. Thank you. We moved part of the information from the manuscript to the Supplemental material as suggested by the Reviewer.

C7. Specific comments

Abstract

Number of subjects not stated.

Line 34 ‘recognized’.

R7. We amended the text as suggested.

C8. Introduction

For a (in my reading) proof-of-concept methodological study, the Introduction is could be shortened a little.  

R8. We thank the Reviewer for the suggestion. As we are targeting a clinical readership as well, we would respectfully ask to keep in the introduction the backgound related to clinical relevance.

C9. Materials and Methods

Line 152 ‘…as follow: ….’  (note colon)

Line 156 ‘ROI’ not defined

Line 155  ‘after radiology had drawn…’

R9. We amended the text as suggested.

C10. The design of Table 1 I find slightly confusing.  The last column is labelled as the r values’, but it seems also to contain P-values.  Perhaps this table could be better designed.  The labelling of columns in Table 2 seems different.

R10. We now edited Table 1 and Table 2 to improve the clarity of reading.

C11. Page 8 line 239 ‘With the CT scan (rather than ‘considering the CT scan’)

R11.  We amended the text as suggested.

C12. Page 9 Paragraph concerning stratifying by SMI.  Was this useful?  Perhaps it could be omitted or (simply the analysis) placed in Supplemental materials, for the more enthusiastic reader.

R12. We think that stratifying by low- and high-SMI may be of clinical relevance as SMI is known to independently associate with outcome in critically ill patients (doi: 10.1016/j.nut.2022.111687). In order to improve the clarity of the paper to the readership of Frontiers in Medicine we moved Table 3 - Comparison of patients with Low versus high Skeletal Muscle Index – to the supplemental material.

C13. Page 11 Line 346 ‘Finally,…’ (rather than ‘At last…’)

R13.  We amended the text as suggested.

Reviewer 2 Report

The aim of this study was to compare diaphragm thickness as evaluated by ultrasound (US) versus computed tomography (CT) during mechanical ventilation. The authors hypothesized that measurements from these two modalities are correlated with each other and that they are related to clinical characteristics. The main findings were that CT and US measurements were correlated, but there were differences in how they relate to sarcopenia. This study presents novel and interesting findings, but a few points should be clarified. 

The skeletal muscle index (SMI) was used as an outcome. Although there was some mention of using CT to assess sarcopenia, it was not clear from the introduction what SMI actually is or why it was included. A brief description of this in the introduction would be helpful to clarify the justification for why CT was acceptable to assess sarcopenia versus other methods (MRI, muscle function, intermuscular fat, etc.). 

In lines 168--170, it was not clear what average measurement was used for analysis. Two US images were acquired from the left and right hemidiaphragms, but was the measurement the average of left and right? Or was there an average for the left and an average for the right? From figure 2, it would seem that only the average of the right side was used, but this was not indicated elsewhere. This should be made clearer considering the correlations that were found. For example, the correlation between US thickness and the left posterior pillar (but not with other areas) appears odd if the US thickness was from the right. If indeed only the left posterior pillar (CT imaging) was correlated with the right costal fibers (US imaging), this should be discussed with possible explanations and implications. 

Perhaps I misunderstood what the different locations of CT images represent. It seems that certain areas on CT should correspond with the costal fibers that were imaged with US at the 9th or 10th intercostal space. Based on this location, the closest corresponding fibers on CT should be the anterior pillars on each side. If this is the case, there should be some discussion on why there were no correlations. 

It is not clear what is meant by "correlation slope is higher than one probably reflects the two structures' different anatomies" in lines 328--331. What is the statistical test being referred to? In addition, unless I am reading the results incorrectly, the left posterior pillar was correlated with US thickness. How does this support the argument that these two areas are distinct? 

The finding that thickness on US was different between those with high vs. low SMI but no difference was found with CT is interesting, but I wonder if this was because of a discrepancy in the timing of assessments. According to the methods used, US images were acquired within 12 hours before or after the CT. As the authors are likely aware, ventilator-induced diaphragm dysfunction happens very quickly. This source of confounding should be discussed. 

In lines 369--372, the authors discuss the possibility that there could be differences in thickness from CT because of the respiratory cycle, but their clinical practice shows that thickness remains stable during CT imaging since there is no active effort on controlled ventilation. Is this consistent with the literature? If not or if the literature is scant, the authors should nonetheless comment on the confounding from respiratory cycles. In addition, even though US thickness was measured with expiration, was the ventilator mode controlled ventilation as well? If not, the authors should also comment on this source of variability. 

The authors allude to the heterogeneity diaphragm thickness based on location. An example of CT images was provided. An example of US imaging of the diaphragm at the zone of apposition would be helpful for comparison. Consider adding this as a figure or as a supplement. 

Author Response

Reviewer #2

C1. The aim of this study was to compare diaphragm thickness as evaluated by ultrasound (US) versus computed tomography (CT) during mechanical ventilation. The authors hypothesized that measurements from these two modalities are correlated with each other and that they are related to clinical characteristics. The main findings were that CT and US measurements were correlated, but there were differences in how they relate to sarcopenia. This study presents novel and interesting findings, but a few points should be clarified.

R1. Thank you very much for the positive feedback and the constructive comments on our manuscript.

C2. The skeletal muscle index (SMI) was used as an outcome. Although there was some mention of using CT to assess sarcopenia, it was not clear from the introduction what SMI actually is or why it was included. A brief description of this in the introduction would be helpful to clarify the justification for why CT was acceptable to assess sarcopenia versus other methods (MRI, muscle function, intermuscular fat, etc.). 

R2. Several imaging modalities may be used to assess sarcopenia, but computed tomography and magnetic resonance are considered the standard reference. Concerning CT imaging, CT-scan derived indexes such as skeletal muscle index (SMI) are correlated with survival among ICU patients (doi: 10.1016/j.nut.2022.111687). Therefore, we used the index itself as an outcome. We detailed this information in the introduction section.

C3. In lines 168--170, it was not clear what average measurement was used for analysis. Two US images were acquired from the left and right hemidiaphragms, but was the measurement the average of left and right? Or was there an average for the left and an average for the right? From figure 2, it would seem that only the average of the right side was used, but this was not indicated elsewhere. This should be made clearer considering the correlations that were found. For example, the correlation between US thickness and the left posterior pillar (but not with other areas) appears odd if the US thickness was from the right. If indeed only the left posterior pillar (CT imaging) was correlated with the right costal fibers (US imaging), this should be discussed with possible explanations and implications.

R3. We considered only US measurements of the right hemidiaphragm. US evaluation of the left hemidiaphragm could not be obtained consistently, it is now explained at the beginning of the results in the paragraph “study population”. Moreover, the study of right hemidiaphragm with US is the benchmark in literature (DOI: 10.1007/s00134-015-3687-3). Considering the aims of the sudy, the primary one is to explore the agreement (evaluated through bias and 95% CI in Bland Altman plots) between the two different techniques (US versus CT). Other objectives aim to investigate how two different techniques - CT scan that is objective and non-operator dependent and US that is operator dependent - are associated and how an US-evaluation of a single area of the diaphragm may be representative of the whole structure of the diaphragm.

C4. Perhaps I misunderstood what the different locations of CT images represent. It seems that certain areas on CT should correspond with the costal fibers that were imaged with US at the 9th or 10th intercostal space. Based on this location, the closest corresponding fibers on CT should be the anterior pillars on each side. If this is the case, there should be some discussion on why there were no correlations.

R4. As described in the manuscript we considered “six different areas: a) right anterior pillar, b) left anterior pillar, c) right posterior pillar, d) left posterior pillar, e) hepatic, and f) splenic domes. The pillars represent the different portions of the diaphragm (the lumbar diaphragm or crura and costal diaphragm), and they can be recognized as bundles of muscular fibers: two arising from the anterolateral surface of the first three right lumbar vertebrae, called “left and right crus of the diaphragm” (where median and lateral arcuate ligaments arise); two anterior parts (between the xiphoid and the middle leaflets of the central diaphragmatic tendon) covering the dome of the liver and spleen; and two costal parts, covering the lateral part of the liver and spleen” in CT scan. On the other hand, US was performed along the ninth or tenth intercostal space near the midaxillary line and angled perpendicular to the chest wall as described by Goligher et al, the so-called “apposition zone”. This area does not exactly belong to a CT area, neither anterior pillars nor the domes.

Diaphragmatic thickness can be adequately measured by CT scan with more reliable results in the posterior pillars’ region. Perhaps this region of the diaphragm can be better assessed and measured with CT in relation to its clear delineation from surrounding structures (i.e., paravertebral muscles, fat tissue). In contrast, the anterior or costal part of the diaphragm is usually more difficult to identify than the lumbar part on transverse images. This may be an explanation for the good correlation between US measurements and the posterior pillar measurements performed with CT scan.

C5. It is not clear what is meant by "correlation slope is higher than one probably reflects the two structures' different anatomies" in lines 328--331. What is the statistical test being referred to? In addition, unless I am reading the results incorrectly, the left posterior pillar was correlated with US thickness. How does this support the argument that these two areas are distinct? 

R5. Correlation slope is referred to the linear regression, it is the angular coefficient of the line of the linear regression. Regarding the second part of the comment, we apologize if we did not clarify the findings enough. We confirm that the primary aim of our study was the agreement between US and CT measurements that we assessed by the Bland-Altman analysis. The right hemidiaphragm US measurements had an excellent agreement – as reported by the Bland-Altman bias – with the hepatic dome and the right anterior pillar assessed by CT. About the analysis of association between US and CT measurements, which was among our secondary aims, the highest correlation between right hemidiaphragm US measurements and the CT measurements was with the left posterior pillar and the right anterior pillar (Table 2).

C6. The finding that thickness on US was different between those with high vs. low SMI but no difference was found with CT is interesting, but I wonder if this was because of a discrepancy in the timing of assessments. According to the methods used, US images were acquired within 12 hours before or after the CT. As the authors are likely aware, ventilator-induced diaphragm dysfunction happens very quickly. This source of confounding should be discussed.

R6. This may be a confounding factor and it is now reported in limitations paragraph of the discussion. However, diaphragmatic thickening seems to happen over days of ICU stay and less frequently over hours (Ref 15: Goligher EC, Fan E, Herridge MS, Murray A, Vorona S, Brace D, Rittayamai N, Lanys A, Tomlinson G, Singh JM, Bolz SS, Rubenfeld GD, Kavanagh BP, Brochard LJ, Ferguson ND: Evolution of Diaphragm Thickness during Mechanical Ventilation. Impact of Inspiratory Effort. Am J Respir Crit Care Med 2015; 192:1080–1088).

C7. In lines 369--372, the authors discuss the possibility that there could be differences in thickness from CT because of the respiratory cycle, but their clinical practice shows that thickness remains stable during CT imaging since there is no active effort on controlled ventilation. Is this consistent with the literature? If not or if the literature is scant, the authors should nonetheless comment on the confounding from respiratory cycles. In addition, even though US thickness was measured with expiration, was the ventilator mode controlled ventilation as well? If not, the authors should also comment on this source of variability.

R7. The effect of respiratory phase on diaphragm thickness during mechanical ventilation in patients receiving neuromuscular blockers (as it is the case of our patients) is negligible. This was demonstrated by Goligher et al. in an article that is now correctly referenced discussion (Ref 14: Goligher EC, Laghi F, Detsky ME, Farias P, Murray A, Brace D, Brochard LJ, Bolz SS, Rubenfeld GD, Kavanagh BP, Ferguson ND: Measuring diaphragm thickness with ultrasound in mechanically ventilated patients: feasibility, reproducibility and validity. Intensive Care Med 2015; 41:642–649). We apologize for the oversight. The ventilator was set in the same way (controlled ventilation, with the same parameters) during CT scan and US evaluation.

C8. The authors allude to the heterogeneity diaphragm thickness based on location. An example of CT images was provided. An example of US imaging of the diaphragm at the zone of apposition would be helpful for comparison. Consider adding this as a figure or as a supplement.

R8. We now proided an exemplary image of US imaging in the supplemental material as Supplemental Figure 1.
